# GenN2N: Generative NeRF2NeRF Translation

## Abstract

We present GenN2N, a unified NeRF-to-NeRF translation framework for various
NeRF translation tasks such as text-driven NeRF editing, super-resolution, object
removal, etc. Unlike previous methods designed for individual translation tasks
with task-specific schemes, GenN2N achieves all these NeRF editing tasks by
employing a universal image-to-image translator to perform editing in the 2D do-
main and lifting 2D edits into the 3D NeRF space. Since the 3D consistency of
2D edits may not be assured, we propose to model the distribution of the underly-
ing 3D edits through a generative model that can cover all possible edited NeRFs.
To model the distribution of 3D edited NeRFs from 2D edited images, we care-
fully design a VAE-GAN that encodes images while decoding NeRFs. The latent
space is trained to align with a Gaussian distribution and the NeRFs are super-
vised through an adversarial loss on its renderings. To ensure the latent code does
not depend on 2D viewpoints but truly reflects the 3D edits, we also regularize
the latent code through a contrastive learning scheme. Extensive experiments on
various editing tasks show GenN2N, as a universal framework, performs as well
or better than task-specific specialists while possessing flexible generative power.

## 1 Introduction

Over the past few years, Neural radiance fields (NeRFs) (Mildenhall et al., 2021) have brought a
promising paradigm in the realm of 3D reconstruction, 3D generation, and novel view synthesis due
to their unparalleled compactness, high quality, and versatility. Extensive research efforts have been
devoted to creating NeRF scenes from 2D images (Melas-Kyriazi et al., 2023; Yu et al., 2021; Cai
et al., 2022; Wang et al., 2022c; Liu et al., 2023) or just text (Poole et al., 2022; Jain et al., 2022)
input. However, once the NeRF scenes have been created, these methods often lack further control
over the generated geometry and appearance. NeRF editing has therefore become a notable research
focus recently.

Existing NeRF editing schemes are usually task-specific. For example, researchers have developed
NeRF-SR (Wang et al., 2022b), OR-NeRF (Yin et al., 2023), NeRF-In (Liu et al., 2022), PaletteN-
eRF (Kuang et al., 2023) for NeRF super-resolution, object removal, inpainting, and color-editing
respectively. These designs require a significant amount of domain knowledge for each specific
task. On the other hand, in the field of 2D image editing, a growing trend is to develop universal
image-to-image translation methods to support versatile image editing (Parmar et al., 2023; Zhang
& Agrawala, 2023; Saharia et al., 2022). By leveraging foundational 2D generative models, e.g.,
stable diffusion (Rombach et al., 2022), these methods achieve impressive editing results without
task-specific customization or tuning. We then ask the question: can we conduct universal NeRF
editing leveraging foundational 2D generative models as well?

The first challenge is the representation gap between NeRFs and 2D images. It is not intuitive how to
leverage image editing tools to edit NeRFs. A recent text-driven NeRF editing method (Haque et al.,
2023) has shed some light on this. The method adopts a "render-edit-aggregate" pipeline. Specif-
ically, it gradually updates a NeRF scene by iteratively rendering multi-view images, conducting
text-driven visual editing on these images, and finally aggregating the edits in the NeRF scene. It
seems that replacing the image editing tool with a universal image-to-image translator could lead
to a universal NeRF editing method. However, the second challenge would then come. Universal
image-to-image translators usually generate diverse and inconsistent edits for different views, e.g.
turning a man into an elf might or might not put a hat on his head, making edits aggregation intricate.
Regarding this challenge, Instruct-NeRF2NeRF (Haque et al., 2023) presents a complex optimiza-

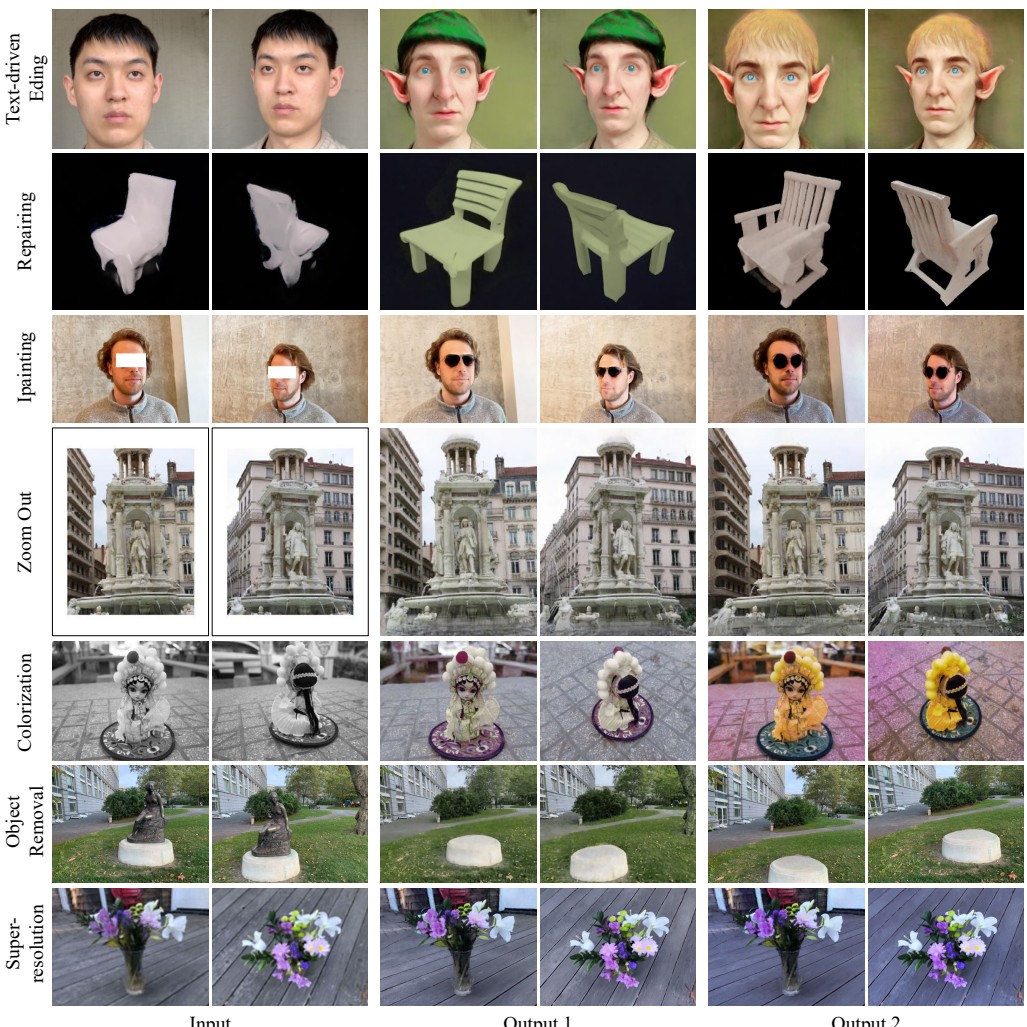

Figure 1: We introduce GenN2N, a unified framework for NeRF-to-NeRF translation, enabling a range of 3D NeRF editing tasks, including text-driven editing, repairing, inpainting, zooming out, colorization, object removal, super-resolution, etc. We show two rendering views of two edited NeRF scenes at inference time (output 1 and output 2). Given a 3D NeRF scene, GenN2N can produce high-quality editing results with suitable multi-view consistency.

tion technique to pursue unblurred NeRF with inconsistent multi-view edits. Due to its complexity, the optimization cannot ensure the quality of the outcomes. Additionally, the unique optimization outcome fails to reflect the stochastic nature of NeRF editing. Users typically anticipate a variety of edited NeRFs just like the diverse edited images.

To tackle the challenges above, we propose GenN2N, a unified NeRF-to-NeRF translation framework for various NeRF editing tasks such as text-driven editing, repairing, inpainting, zoom out, colorization, object removal, super-resolution (see Fig. 1). In contrast to Instruct-NeRF2NeRF, GenN2N adopts a "render-edit-generate" pipeline. We first render a NeRF scene into multi-view images, then exploit a universal image-to-image translator to edit different views, and finally learn a generative model to depict the distribution of NeRF edits. Instead of aggregating all the image edits to form a single NeRF edit, our key idea is to embrace the stochastic nature of content editing by modeling the distribution of the edits in the 3D NeRF space.

Specifically given a NeRF model or its multi-view images, along with the editing goal, we first generate edited multi-view images using a universal image-to-image translator. Each view corresponds to a unique 3D edit with some geometry or appearance variations. Conditioned on the input NeRF,

GenN2N trains a conditional 3D generative model to reflect such content variations. At the core of GenN2N, we design a 3D VAE-GAN that incorporates a differentiable volume renderer to connect 3D content creation with 2D GAN losses, ensuring that the inconsistent multi-view renderings can still help each other regarding 3D generation. Moreover, we introduce a contrastive learning loss to ensure that the 3D content variation can be successfully understood just from edited 2D images without being influenced by the camera viewpoints. During inference, users can simply sample from the conditional generative model to obtain various 3D editing results aligned with the editing goal.

We have conducted experiments on human, items, and 360-degree environment scenes for various editing tasks such as text-driven editing, repairing, inpainting, zoom out, colorization, object removal, and super-resolution, demonstrating the effectiveness of GenN2N in supporting diverse NeRF editing tasks while keeping the multi-view consistency of the edited NeRF.

We summarize the contribution of this paper as follows,

- A generative NeRF-to-NeRF translation formulation for the universal NeRF editing task together with a generic solution;
- a 3D VAE-GAN framework that can learn the distribution of all possible 3D NeRF edits corresponding to the a set of input edited 2D images;
- a contrastive learning framework that can disentangle the 3D edits and 2D camera views from edited images;
- extensive experiments demonstrating the superior efficiency, quality, and diversity of the NeRF-to-NeRF translation results.

## 2 RELATED WORK

**NeRF Editing.** Previous works such as EditNeRF (Liu et al., 2021) propose a conditional neural field that enables shape and appearance editing in the latent space. PaletteNeRF (Kuang et al., 2022; Wu et al., 2022) focuses on controlling color palette weights to manipulate appearance. Other approaches utilize bounding boxes (Zhang et al., 2021), meshes (Yuan et al., 2022), point clouds (Chen et al., 2023), key points (Zheng et al., 2022), or feature volumes (Lazova et al., 2023) to directly manipulate the spatial representation of NeRF. However, these methods either heavily rely on user interactions or have limitations in terms of spatial deformation and color transfer capabilities.

**NeRF Stylization.** Images-referenced stylization (Huang et al., 2022; Chiang et al., 2022; Zhang et al., 2022) often prioritize capturing texture style rather than detailed content, resulting in imprecise editing appearance of NeRF only. Text-guided works (Wang et al., 2023; 2022a), on the other hand, apply contrastive losses based on CLIP (Radford et al., 2021) to achieve the desired edits. While text references usually describe the global characteristics of the edited results, instructions offer a more convenient and precise expression.

**Instruct-driven NeRF editing.** Among numerous image-to-image translation works, Instruct-Pix2Pix (Brooks et al., 2022) stands out by efficiently editing images following instructions. It leverages large pre-trained models in the language and image domains (Brown et al., 2020; Rombach et al., 2022) to generate paired data (before and after editing) for training. While editing NeRF solely based on edited images is problematic due to multi-view inconsistency. To address this, an intuitive yet heavy approach (Haque et al., 2023) is to iteratively edit the image and optimize NeRF. Inspired by Generative Radiance Fields (Schwarz et al., 2020; Chan et al., 2022), We capture various possible NeRF editing in the generative space to solve it.

## 3 METHOD

Given a NeRF scene, we present a unified framework GenN2N to achieve various editing on the 3D scene like in the 2D image editing domain, such as text-driven editing, zoom out, inpainting, colorization, super-resolution, object removal, etc. Here, we formulate those 2D image editing methods as a universal image-to-image translator and those NeRF editing tasks as the NeRF-to-NeRF translation task, in which the given NeRF is translated into NeRF scenes with high rendering quality and 3D geometry consistency according to the user-selected editing target. The overview of GenN2N

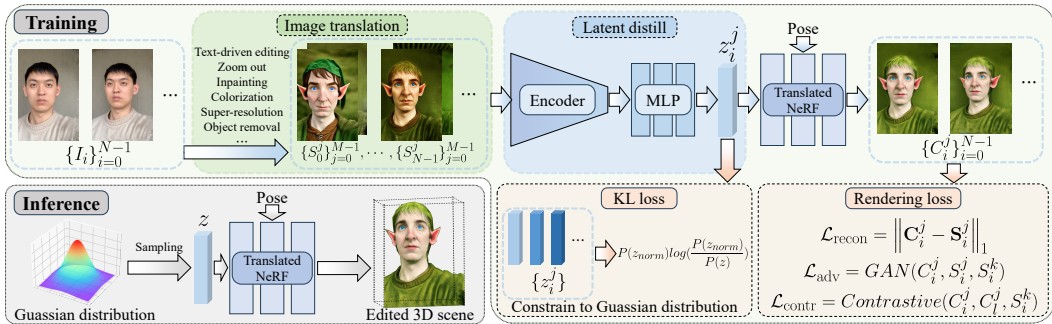

Figure 2: Overview of GenN2N. We first edit the source image set $\{I_i\}_{i=0}^{N-1}$ using 2D image-to-image translation methods, e.g., text-driven editing, colorization, zoom out, etc. For each view $i \in [0, N-1]$, we generate $M$ edited images, resulting in a group of translated image set $\{\{S_i^j\}_{j=0}^{M-1}\}_{i=0}^{N-1}$. Then we use the Latent Distill Module to learn $M \times N$ edit code vectors from the translated image set, which serve as the input of the translated NeRF. To optimize our GenN2N, we design four loss functions: a KL loss to constrain the latent vectors to a Gaussian distribution; and $\mathcal{L}_{\text{recon}}$, $\mathcal{L}_{\text{adv}}$ and $\mathcal{L}_{\text{contr}}$ to optimize the appearance and geometry of the translated NeRF. In the inference stage, we can sample a latent vector $z$ from Gaussian distribution and render a corresponding multi-view consistent 3D scene with high quality.

is illustrated in Fig. 2, we first perform image-to-image translation in the 2D domain leveraging foundational 2D generative models (Rombach et al., 2022), and then lift 2D edits to 3D and achieve NeRF-to-NeRF translation. Given N multi-view images $\{I_i\}_{i=0}^{N-1}$, we first use Nerfstudio (Tancik et al., 2023) to train the original NeRF. Then we use a universal image-to-image translator to edit these source images. However, the content generated by the 2D translator may be inconsistent among multi-view images. For example, using different initial noise, the 2D translator (Avrahami et al., 2023) may generate different content for image inpainting, which makes it difficult to ensure the 3D consistency between different view directions in the 3D scene. To ensure the 3D consistency and rendering quality, we propose to model the distribution of the underlying 3D edits through a generative model that can cover all possible edited NeRFs, by learning an edit code for each edited image so that the generated content can be controlled by this edit code during the NeRF-to-NeRF translation process.

For each view $\in [0, N-1]$, we generate M edited images, resulting in a group of the translated image set $\{\{S_i^j\}_{j=0}^{M-1}\}_{i=0}^{N-1}$. Then we design a Latent Distill Module in Sec. 3.1 to map each translated image $S_i^j$ into an edit code vector $z_i^j$ and design a KL loss $\mathcal{L}_{\text{KL}}$ to constrain those edit code vectors to a Gaussian distribution. Conditioned on the edit code $z_i^j$, we perform NeRF-to-NeRF translation in Sec. 3.2 by rendering multi-view images $\{C_i\}_{i=0}^{N-1}$ and optimize the translated NeRF by three loss functions: the reconstruction loss $\mathcal{L}_{\text{recon}}$, the adversarial loss $\mathcal{L}_{\text{AD}}$, and the contrastive loss $\mathcal{L}_{\text{contr}}$. After the optimization of the translated NeRF, as described in Sec. 3.3, we can sample an edit code z from Gaussian distribution and render the corresponding edited 3D scene with high quality and multi-view consistency in the inference stage.

## 3.1 LATENT DISTILL MODULE

**Image Translation.** As illustrated in Fig. 2, GenN2N is a unified framework for NeRF-to-NeRF translation, in which the core is to perform a universal 2D image-to-image translation and lift 2D edits into 3D NeRF-to-NeRF translation. Given the source multi-view image set $\{I_i\}_{i=0}^{N-1}$ of a NeRF scene, we first perform image editing $MN$ times for each view using a universal 2D image-to-image translator, producing a group of translated image set $\{\{S_i^j\}_{j=0}^{M-1}\}_{i=0}^{N-1}$. Here, we denote the universal 2D image-to-image translator as a generative model that can solve arbitrary 2D image editing tasks. In this paper, we use several 2D translation tasks to show the adaptability of our GenN2N: text-driven editing, zoom out, inpainting, colorization, super-resolution, and object removal. For more details about those 2D image editing methods, please refer to the supplementary materials.

**Edit Code.** Since 2D image-to-image translation may generate different content for different edits, causing the inconsistency problem in the 3D scene. We propose to map each edited image $S_i^j$ into a

latent feature vector named edit code to characterize these diverse editings. We employ the off-the-shelf VAE encoder from stable diffusion (Rombach et al., 2022) to extract the feature from $S_i^j$ and then apply a tiny MLP network to produce this edit code $z_i^j \in \mathbb{R}^{64}$. During the training process, we keep the pre-trained encoder fixed and only optimize the parameters of the tiny MLP network. This mapping process can be formulated as follows:

$$z_i^j = \mathcal{D}(S_i^j) = \mathcal{M}(\mathcal{E}(S_i^j)) \tag{1}$$

where $\mathcal{D}$ represent this mapping process, $\mathcal{E}$ is the fixed encoder, and $\mathcal{M}$ is the learnable tiny MLP.

**KL loss.** In order to facilitate effective sampling of the edit code so as to control the editing diversity of our NeRF-to-NeRF translation, we need to constrain the edit code to a well-defined distribution. Thus we design a KL loss to encourage $z_i^j$ to approximate a Gaussian distribution:

$$\mathcal{L}_{\text{KL}} = \mathbb{E}_{S \in \{\{S_i^j\}_{j=0}^{M-1}\}_{i=0}^{N-1}} [P(z_{normal}) log(\frac{P(z_{normal})}{P(\mathcal{D}(S))})] \tag{2}$$

where $P(z_{normal})$ denotes probability distribution of the standard Gaussian distribution in $\mathbb{R}^{64}$ and $P(\mathcal{D}(S))$ means probability distribution of the extracted edit codes.

**Contrastive loss.** It is not assured that edit codes $\mathbf{z}$ obtained from the Latent Distillation Module contain only the editing information while excluding viewpoint-related effects. However, since the translated NeRF utilizes $\mathbf{z}$ to edit scenes, it yields instability if $\mathbf{z}$ violently changes given images that are similar in appearance but different in viewpoints. To ensure the latent code does not depend on 2D viewpoints but truly reflects the 3D edits, we regularize the latent code through a contrastive learning scheme. Specifically, we reduce the distance between edit codes of different-view rendered images from a translated NeRF that share the same edit code, while increasing the distance between same-view images that are multi-time edited by the 2D image-to-image translator. Given an edit code $z_i^j$ extracted from the $i$-th input view at the $j$-th edited image $S_i^j$, we render multi-view images $\{C_i^j\}_{i=0}^{N-1}$ using the translated NeRF conditioned on $z_i^j$. Then we employ contrastive learning to encourage the edit code $z_i^j$ to be close to $\{\acute{z}_l^j\}_{l=0}^{N-1}$ extracted from $\{C_l^j\}_{l=0}^{N-1}$, where $l \neq i$ , while being distinct from the edit codes $\{z_i^k\}_{k=0}^{M-1}$ extracted from $\{S_i^k\}_{k=0}^{M-1}$, where $k \neq j$.

Specifically, our contrastive loss is designed as follows:

$$\begin{aligned}
\mathcal{L}_{\text{contr}} &= \mathcal{L}_{\text{contr}}^{\text{att}} + \mathcal{L}_{\text{contr}}^{\text{rep}} \\
&= \sum_{l=0}^{N-1} ||z_i^j - \acute{z}_l^j||_2^2 + \sum_{k=0}^{M-1} max(0, \alpha - ||z_i^j - z_i^k||_2^2), l \neq i \text{ and } k \neq j
\end{aligned} \tag{3}$$

where $\alpha$ represents the margin that encourages the difference in features.

## 3.2 NeRF-to-NeRF translation

**Translated NeRF.** After 2D image-to-image translation, we need to lift these 2D edits to the 3D NeRF. For this purpose, we propose to modify the original NeRF as a translated NeRF that takes the edit code $z$ as input and generates the translated 3D scene according to the edit code. We refer readers to the supplementary for more details about the network architecture.

**Reconstruction loss.** Given an edit code $z_i^j$ extracted from the edited image $S_i^j$, we can generate a translated NeRF to render $C_i^j$ from the same viewpoint. Then we define the reconstruction loss as the L1 normalization and Learned Perceptual Image Patch Similarity (LPIPS) (Zhang et al., 2018) between the rendered image $C_i^j$ and the edited image $S_i^j$ as follows:

$$\mathcal{L}_{\text{recon}} = \mathcal{L}_{\text{L1}} + \mathcal{L}_{\text{LPIPS}} = \left\| \mathbf{C}_i^j - \mathbf{S}_i^j \right\|_1 + LPIPS[\mathcal{P}(\mathbf{C}_i^j) - \mathcal{P}(\mathbf{S}_i^j)] \tag{4}$$

where $\mathcal{P}$ means a patch sampled from the image. Note that due to the lack of 3D consistency of the edited multi-view image, the supervision of the edited image from other viewpoints $\{S_l^j\}_{l \neq i}$ will lead to conflicts in pixel-space optimization. Therefore, we only optimize the translated NeRF using the same view image $S_i^j$.

**Adversarial loss.** Since the 3D consistency of edited multi-view images is not assured, relying solely on the reconstruction loss on the same view often leads to blurry results. Previous research demonstrates the effectiveness of adversarial training in preventing the production of blurry rendered images resulting from conflicts that arise from noise in the camera extrinsic when performing image supervision from different viewpoints (Huang et al., 2020).

To encourage high-quality output and address artifacts caused by inconsistent cross-view translated images, we incorporate adversarial loss on rendered images from the translated NeRF. Instead of directly differentiating real and fake samples, we learn the relation of the pair of samples by using another edited images with the same viewpoints as conditions for the discriminator and take the difference between the target image and generated images as input. This design is conducive for the discriminator to distinguish similarities between condition images and target images, thus promoting the content of the translated NeRF to be consistent with the purpose of 2D image editing.

Specifically, the discriminator $\mathbf{D}$ takes into real pairs and fake pairs. Each real pair $\mathbf{R}$ consists of $S^j$ and $S^j - S^k$ where $S^j \in \{S_i^j\}_{i=0}^{N-1}$ and $S^k \in \{S_i^k\}_{i=0}^{N-1}$ are from two sets of edited images from a universal image translator. Similarly, each fake pair $\mathbf{F}$ consists of $C^j$ and $C^j - S^k$ in which $C^j \in \{C_i^j\}_{i=0}^{N-1}$ is generated by translated NeRF. Note that the images in the same pair come from the same viewpoint. The pairs are concatenated in RGB channels and fed into the discriminator. We optimize the discriminator $\mathbf{D}$ and translated NeRF with the objective functions below:

$$\begin{aligned} \mathcal{L}_{\text{AD-D}} &= \mathbb{E}_R[-log(D(R))] + \mathbb{E}_F[-log(1 - D(F))] \\ \mathcal{L}_{\text{AD-G}} &= \mathbb{E}_F[-log(D(F))] \end{aligned} \tag{5}$$

**Optimization.** During the training process, we jointly optimize the loss functions mentioned above: $\mathcal{L}_{\text{KL}}$ and $\mathcal{L}_{\text{contr}}$ for the edit code, $\mathcal{L}_{\text{recon}}$ and $\mathcal{L}_{\text{AD-G}}$ for the translated NeRF, and $\mathcal{L}_{\text{AD-D}}$ for the discriminator. The total loss formula is expressed as follows:

$$\mathcal{L} = \mathcal{L}_{\text{KL}} + \mathcal{L}_{\text{recon}} + \mathcal{L}_{\text{AD-G}} + \mathcal{L}_{\text{AD-D}} + \mathcal{L}_{\text{contr}} \tag{6}$$

where we assign each regularization term the weight of 1.0, 1.0, 0.1, 0.1, 0.1 in all of our experiments. The weights can be adjusted to prioritize different aspects of the training objective, such as reconstruction accuracy, adversarial training, and perceptual quality.

### 3.3 INFERENCE

After the optimization of our GenN2N, the translated NeRF is optimized to be able to render the target scene conditioned on the edit code. As shown in Fig. 1, users can simply sample an edit code from the Gaussian distribution and use the translated NeRF to render the 3D scene with high-quality and multi-view 3D consistency.

## 4 EXPERIMENTS

Our proposed GenN2N is a unified NeRF-to-NeRF translation framework which can support various NeRF editing tasks. In this paper, we demonstrate the effectiveness of GenN2N by a suite of challenging NeRF-to-NeRF translation tasks:

(1) **Text-driven Editing** edits the given NeRF scene to a diversity of NeRF scenes according to the input text instruction.
(2) **Super-resolution** enhances the resolution of NeRF and enables multiple plausible outcomes.
(3) **Object Removal** removes the target object in the NeRF scene while keeping other contents, especially the background content, unchanged and plausible.
(4) **Zoom Out** extends an input NeRF along the input region to enlarge NeRF scenes.
(5) **Inpainting** fills in user-specified masked regions in the NeRF scene with realistic content.
(6) **Colorization** transforms a gray-scale NeRF scene to a set of plausible color NeRF scenes.

We achieve those tasks by simply changing the 2D image-to-image translator in our framework, without any additional task-specific design. Previous studies have extensively explored some of these issues like text-driven editing, super-resolution, and object Removal. However, there is rarely a unified framework that can achieve all these problems with strong performance, high quality, and plausible multi-view consistent 3D structure. Furthermore, GenN2N can also perform zooming

out, inpainting, and colorization in NeRF-to-NeRF translation, which were not explored in prior research. We refer readers to the supplementary materials for detailed experiment settings, dataset settings and implementation details.

## 4.1 COMPARISONS

**Text-drive Editing.** We achieve text-driven editing of the given NeRF by using Instruct-Pix2Pix (Brooks et al., 2022) as the 2D image-to-image translator in our framework. Instruct-Pix2Pix can efficiently edit images following user instructions, which causes the the 3D inconsistency problem between different edits. While Instruct-NeRF2NeRF (Haque et al., 2023) proposed an iterative updating mechanism to address this issue, it falls short in modeling the diversity of different edits, making it challenging to ensure the quality of the outcomes. We conduct experiments on Face (Haque et al., 2023) and Fangzhou self-portrait (Wang et al., 2023) dataset to compare GenN2N with Instruct-NeRF2NeRF. Quantitative results are shown in Table 1, where we use CLIP Text-Image Direction Similarity, CLIP Direction Consistency and Fréchet Inception Distance (FID) (Heusel et al., 2017) as the evaluation metric. The results highlight the superior performance of GenN2N over Instruct-NeRF2NeRF, demonstrating its effectiveness. Additionally, qualitative comparison results in Fig. 3 provide further insights. Since the multi-view inconsistency caused by the 2D image-to-image translator, Instruct-NeRF2NeRF has conflicting optimization goals for different viewpoints when editing NeRF. Therefore, in many cases, it can not achieve the desired editing effect as GenN2N, like it fails to remove the vase on the desk in Fig. 3 the bottom two rows.

**Super-resolution.** When only low-resolution images are available, our methods can boost NeRF in reconstructing scenes at higher resolution, while keeping view consistency and avoiding blurry outputs. We achieve this by employing ResShift (Yue et al., 2023) as the 2D image-to-image translator in GenN2N. Following NeRF-SR (Wang et al., 2022b), we conduct experiments on LLFF dataset (Mildenhall et al., 2021), using PSNR and SSIM as evaluation metrics. As shown in Table 2, GenN2N obtains NeRF-to-NeRF translation with higher performance than NeRF-SR. Moreover, we also provide qualitative comparison results in Fig. 3, where GenN2N produces clearer and more realistic rendering results than NeRF-SR.

**Object Removal.** The goal of NeRF object removal is to remove objects from the NeRF scene, guided by user-provided points or text prompts. OR-NeRF (Yin et al., 2023) achieves this through a multi-step process: it employs SAM (Kirillov et al., 2023) for object segmentation, utilizes Blended Latent Diffusion (Avrahami et al., 2023) to fill in the background content in multi-view images, and subsequently trains the NeRF model with color, depth, and perceptual cues. In our experiments, we use SAM and LaMa as the 2D image-to-image translator in our GenN2N, which is the same setting as OR-NeRF. Quantitative comparisons on SPIn-NeRF (Mirzaei et al., 2023) dataset are shown in Table 3, where GenN2Nachieves superior PSNR and SSIM scores than OR-NeRF, highlighting the effectiveness of our GenN2N framework. In addition, qualitative results are showcased in Fig. 3 revealing that while SPIn-NeRF fails to generate reasonable content behind the removed object, while our GenN2N produces realistic content in the same region with fine multi-view consistency.

## 4.2 APPLICATIONS

**Zoom Out.** Given a NeRF scene optimized from multi-view images with a limited field-of-view, GenN2N can enlarge the NeRF scene by leveraging Blended Latent Diffusion (Avrahami et al., 2023) as the 2D image-to-image translator. Notably, this translation task has not been explored by previous methods, thus we only provide qualitative results in Fig. 1 and appendix. For each scene, we show two source images captured from different viewpoints, along with their corresponding translated rendering results using different edit codes. As can be seen, our method can produce reasonable content in the expanded regions.

**Inpainting.** NeRF inpainting is to fill in the 3D content of regions specified by users. We achieve 3D NeRF inpainting by using Blended Latent Diffusion (Avrahami et al., 2023) as the 2D translator to inpaint those masked regions in the 2D domain. Qualitative results are shown in Fig. 1, where we show the translated results from multiple viewpoints to demonstrate that reasonable and high-quality content can be generated with harmonious multi-view consistency.

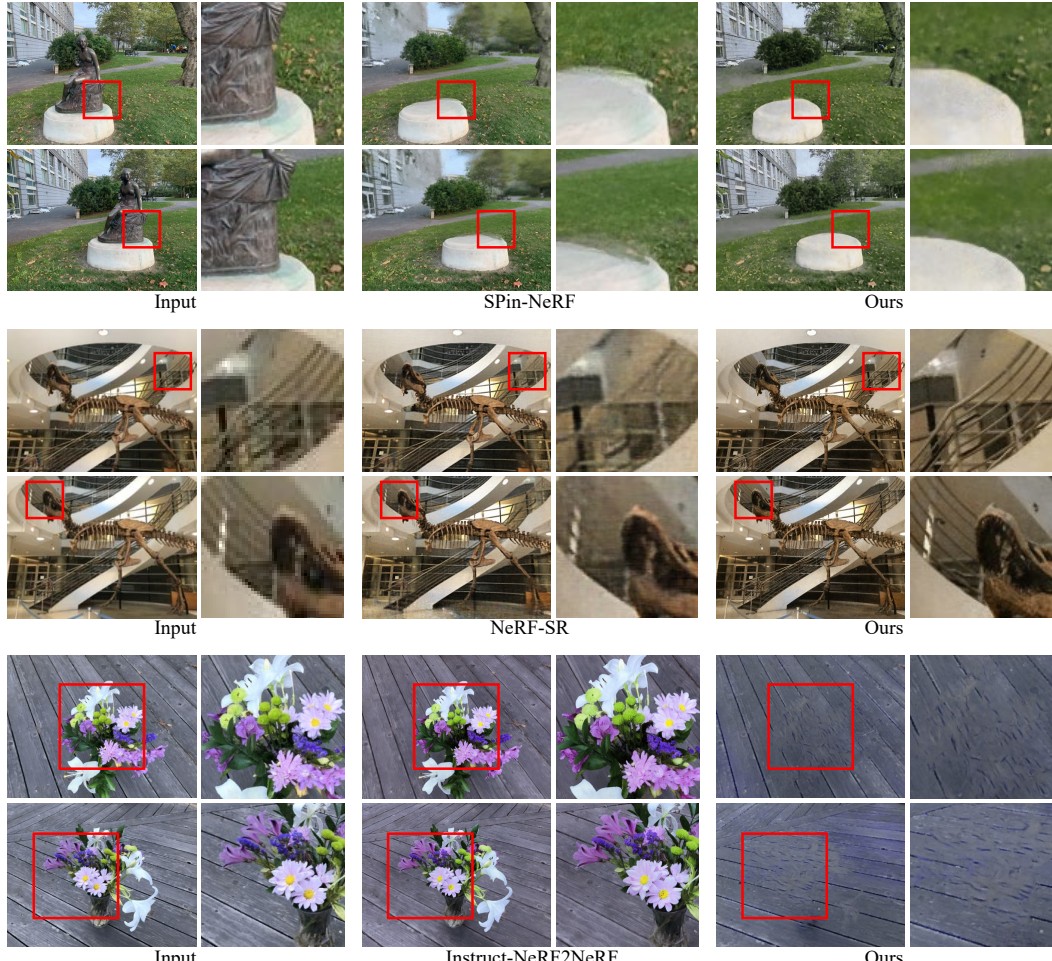

Figure 3: **Comparisons with baselines.** We compare with SPIn-NeRF (Mirzaei et al., 2023)in object removal on the data provided by SPIn-NeRF. Our method preserves details of the scene while successfully eliminating the object, while SPInNeRF (Mirzaei et al., 2023) causes the surrounding scene to blur and deform. We compare with NeRF-SR (Wang et al., 2022b) in the super-resolution shown in middle two rows. And compare with Instruct-NeRF2NeRF (Haque et al., 2023) in the editing by using the text prompt "remove the vase" in the bottom two rows.

Table 1: Text-driving editing results.

| Method | CLIP Text-Image Direction Similarity ↑ | CLIP Direction Consistency ↑ | FID ↓ |
|---|---|---|---|
| Instruct-N2N | 0.0728 | 0.9196 | 781.31 |
| Ours | **0.0794** | **0.9379** | **424.44** |

Table 2: Super-resolution results.

| Method | PSNR ↑ | SSIM ↑ | LPIPS ↓ |
|---|---|---|---|
| NeRF-SR | 27.957 | 0.897 | 0.0937 |
| Ours | **28.501** | **0.913** | **0.0748** |

**Colorization.** For colorization, we use DDColor (Kang et al., 2022) as the 2D image-to-image translator in our GenN2N to translate a gray-scale NeRF scene into a colored 3D scene. We show initial gray-scale images and our translated results in Fig. 1 and appendix. We can find that with different edit codes, the scene is rendered in different color styles. It is noticeable that with the same edit code, the color rendered from different views is consistent. This strongly demonstrates the effectiveness of our method in translating NeRF while keeping the 3D consistency of the scene.

## 4.3 ABLATION STUDIES

We conduct comprehensive ablation experiments to validate the designs of each component in our method. Due to space limitations, we only highlight the essential aspects of GenN2N below.

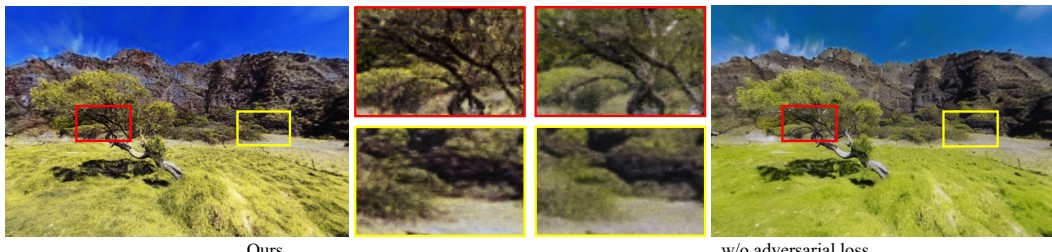

Ours        w/o adversarial loss

Figure 4: **Ablations.** The removal of our adversarial loss results in blurry novel view images with artifacts, especially in the zoom in regions.

Table 3: Object removal results.

| Method | PSNR ↑ | SSIM ↑ | LPIPS ↓ |
|--------|--------|--------|---------|
| SPin-NeRF | 22.529 | 0.621 | **0.317** |
| Ours | **24.235** | **0.654** | 0.326 |

Table 4: Ablation study.

| Method | CLIP Text-Image Direction Similarity ↑ | CLIP Direction Consistency ↑ |
|--------|---------------------------------------|------------------------------|
| w/o Contrastive loss | 0.0460 | **0.9650** |
| w/o Adversarial loss | 0.0646 | 0.9642 |
| Ours | **0.0794** | 0.9379 |

**The Contrastive Loss.** We demonstrate the advantages of incorporating our proposed contrastive loss in Table 4. The motivation is to disentangle the pose and edit information present in the latent space. We achieve this by reducing the distance between edit codes of different-view rendered images from a translated NeRF that shares the same edit code, while increasing the distance between same-view images that are edited by the 2D image-to-image translator with different edit codes. As demonstrated in Table 4, the absence of contrastive loss leads to the generation of blurry areas in the rendered images, resulting in a decrease in the metric scores. This blurriness can be attributed to the inclusion of pose information within the edit code $z$. By incorporating the contrastive loss, our method successfully achieves a uniform appearance with different observing views under the same style latent $z$.

**Discriminator for Novel Views.** We demonstrate the effectiveness of employing a conditional discriminator to address artifacts caused by inconsistent cross-view edited images and to enhance the quality of novel view rendering images, as depicted in Fig. 4. The removal of this conditional discriminator results in blurry novel view images with artifacts in the background region. We attribute these undesirable effects to the inability of current image-to-image translation methods, such as InstructPix2Pix, to produce image editing consistently across multi-view images. To mitigate these issues, we introduce a conditional discriminator between rendered images from the translated NeRF and edited images from the 2D image-to-image translator. This inclusion successfully eliminates artifacts and enhances the image quality of rendered images from the translated NeRF.

## 5 CONCLUSIONS

We have presented GenN2N, a unified NeRF-to-NeRF translation framework designed to address various NeRF editing tasks. Unlike previous methods that often relied on task-specific approaches, our framework leverages a universal image-to-image translator to perform editing in the 2D domain and integrates 2D edits into 3D NeRF space. As 2D editing often exhibits variations in the generated content, making it difficult to ensure the 3D consistency of the translated NeRF, we propose to model the distribution of 3D edited NeRFs from 2D edited images. Also, we designed several techniques including the latent distill module, the KL loss, the reconstruction loss, the adversarial loss, and the contrastive loss. After the optimization of GenN2N, users can simply sample from the conditional generative model to obtain diverse 3D editing results with multi-view consistency and high rendering quality. We have conducted comprehensive experiments to show that GenN2Ncan produce superior efficiency, quality, and diversity compared with existing task-specific methods on various editing tasks including text-driven editing, super-resolution, object removal, zoom out, inpainting, and colorization.

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
