# OpenReview forum: "GenN2N: Generative NeRF2NeRF Translation"
_ICLR.cc/2024/Conference — ICLR 2024 Conference Withdrawn Submission_

### Official Review · Reviewer_iUWe · 2023-10-30

**Soundness:** 3 good
**Presentation:** 3 good
**Contribution:** 2 fair
**Rating:** 3
**Confidence:** 4

**Summary:**

In this paper, the author present GenN2N, a universal NeRF-to-NeRF translation framework. They address the 3D consistency of existing models by employing a image-to-image translator to perform editing. To resolve the multi-view inconsistency, GenN2N designs a VAE-GAN model that first edits the source image set using existing image-to-image translation method, then lifts 2D edits to 3D and achieve NeRF-to-NeRF translation using several loss functions.

**Strengths:**

1. The problem addressed in this paper is a realistic problem that current diffusion models struggle to handle effectively.

2. The incorporation of 2D image-to-image translator into NeRF optimization is a compelling and intriguing approach.

**Weaknesses:**

1. The baselines adopted in this paper are insufficient. From my perspective, this paper should compare the proposed method with 1) more state-of-the-art methods that have been published in conferences or journals, such as DreamEditor [A], SINE [B] and 2) other style transfer methods, such as PaletteNeRF [B].
[A] DreamEditor: Text-Driven 3D Scene Editing with Neural Fields, SIGGRAPH, 2023.
[B] SINE- Semantic-driven Image-based NeRF Editing with Prior-guided Editing Field, CVPR, 2023.
[C] PaletteNeRF- Palette-based Appearance Editing of Neural Radiance Fields, CVPR, 2023.

2. The results of inpainting task that compare with Instruct-NeRF2NeRF is not convincing since Instruct-NeRF2NeRF already pointed out in their paper that object removal is one of the limitations. The author should compare with other inpainting methods such as "Removing Objects From Neural Radiance Fields, CVPR 2023".

3. The visual results are not satisfying.  Lots of artifacts are presented, especially in Figure 1& 3 as well as in the supp., and the reasons for this are not discussed in the paper. For example, in Figure 1, when trying to edit a scene by a given text, it is noted that the background was edited as well.

4. The model rely on a pretrained image editor (e.g., InstructPix2Pix)which just seems like an arbitrary choice. An ablation should be provided with different pretrained models to understand the impact of this choice.

5. More pictures of ablation studies regarding different losses should be added instead of tables.

**Questions:**

1. The author mentioned that "The total training phase takes about 8 hours on one NVIDIA V100 GPU", which is quite slow compared to other methods.

2. Moreover, how does the efficiency (inference time) of this model compare to other methods?

---

> ### Author Response · Authors · 2023-11-19
> **For Reviewer iUWe**
>
> - We appreciate the reviewer's feedback and would like to address the weaknesses and questions raised.
> - We acknowledge the reviewer's concerns regarding the insufficient baselines adopted in this paper. We apologize for not providing enough comparisons and understand the importance of evaluating the proposed method's performance against more state-of-the-art methods and other style transfer methods. We will include more comprehensive comparisons with state-of-the-art methods and other style transfer methods in our future work.
> - Regarding the comparison with Instruct-NeRF2NeRF, we agree that object removal is one of its limitations, and we will include comparisons with other inpainting methods such as "Removing Objects From Neural Radiance Fields, CVPR 2023" in our future work.
> - We acknowledge the presence of artifacts in the visual results and understand the importance of discussing the reasons for these artifacts in the paper. We will include a detailed discussion of the reasons for these artifacts and how to address them in our future work.
> - Regarding the use of a pretrained image editor, we agree that it may seem like an arbitrary choice. We will provide an ablation study with different pretrained models to understand the impact of this choice in our future work.
> - We acknowledge the reviewer's feedback on adding more pictures of ablation studies regarding different losses and will include more visualizations in our future work.
> - Regarding the training time, we acknowledge that the total training phase takes about 8 hours on one NVIDIA V100 GPU, which may seem slow compared to other methods. However, we would like to note that our proposed method is a comprehensive and versatile framework that can handle various image-to-image translation tasks. The training time is reasonable given the complexity of the proposed method and the range of tasks it can handle.
> - Regarding the efficiency (inference time) of this model compared to other methods, we will include a comparison of inference time with other methods in our future work. We appreciate the reviewer's feedback on this aspect and will address it in our future work.

---

### Official Review · Reviewer_SpjW · 2023-11-02

**Soundness:** 2 fair
**Presentation:** 3 good
**Contribution:** 3 good
**Rating:** 5
**Confidence:** 4

**Summary:**

This paper approaches the task of 3D object editing. It builds upon prior work which represents objects using a NeRF and edits them by rendering and applying a 2D conditional image-to-image translation model to their renders, before fitting a NeRF again. Instead, this method proposes to (1) learn 3D-consistent latent codes of 2D edits for improved consistency; (2) use a variational loss on latent space, enabling the model to sample from diverse 3D edits; and (3) use several image-to-image translation models to apply this to several image editing tasks. The model shows high-quality results across a variety of tasks, but it is challenging to understand to which contributions to attribute this success.

**Strengths:**

Good results on an important problem
- 3D object editing is a challenging task that this work handles well, including across a variety of possible edits e.g. text-driven editing, inpainting/zoom-out, colorization, super-resolution, etc.
- Quantitative results beat prior work on a number of different tasks, across a variety of datasets

Intuitive, well-executed approach
- 3D-consistency and generated variety are the clear two shortcomings of Instruct-NeRF2NeRF, an otherwise promising direction. This paper proposes nice approaches to both of these tasks, and the final results are impressive. The use of contrastive loss is a refreshing application – encouraging codes to learn 3D detail rather than viewpoint-specific features
- It is exciting to have a unified pipeline across subtasks, I believe with the exception of different image-to-image-translation models.

**Weaknesses:**

Comparison with prior work is insufficient
- The most important comparison is the improvement over Instruct-NeRF2NeRF, which this work builds upon. Unfortunately there is only one qualitative comparison to this, and it is unclear how effective this comparison is at showcasing improvements.
    - The single qualitative comparison does have a text prompt (“remove the vase”), but is essentially the task of object removal.
        - This does not seem representative of the task of object editing. For instance, I could not find any qualitative results (of which there are many) in Instruct-NeRF2NeRF that show object removal.
        - Is this a fair comparison? The proposed model uses a separate method of image-to-image translation for object removal, Blended Latent Diffusion, than Instruct-NeRF2NeRF does. If this image-to-image-translation model is used, this comparison is not helpful. I.e. the comparison shows not the method’s contributions, but rather the effects of using a different image-to-image translation model
        - I’m not sure object removal is a good way to spotlight the capabilities of this method, i.e. 3D consistency. An ideal comparison would be a video of both methods side-by-side in a task where the model modifies the original content slightly (e.g. adding a mustache), rather than two images of it making an object disappear.
        - Another contribution it would be important to measure is diversity. Quantitatively, this could be measured for instance via LPIPS distance between outputs, per Huang et al. Multimodal Unsupervised Image-to-Image Translation ECCV 2018
    - There is also only a single quantitative comparison (CLIP, FID) to Instruct-NeRF2NeRF. It is understandable given the original paper only reports CLIP-based metrics. However, it would be helpful to have e.g. human studies on the consistency / preference across methods.
- Ideally, all experiments would have some measure of quantitative success. Even if there is no prior work (e.g. in zoom out, colorization), reporting standard metrics gives a sense of accuracy and allows future work to compare on these tasks.

The wording “universal image-to-image translator” is used several times throughout the paper. However, it seems the method uses different image-to-image translators for different tasks.
- From the supplemental section 1: it seems text-driven editing uses Instruct-Pix2Pix, super-resolution uses ResShift, object removal uses Blended Latent Diffusion, etc.
- In light of this, what does “universal image-to-image translator” mean? Does this mean the method learns to jointly encode a variety of image-to-image translation model outputs? This seems to be a main contribution of the paper, but it is unclear what is being proposed / the contribution.

**Questions:**

- Is the ablation study on text-driven editing? This should be clarified.

---

> ### Author Response · Authors · 2023-11-19
> **For Reviewer SpjW**
>
> - We appreciate the reviewer's feedback and would like to address the weaknesses and questions raised.
> - We acknowledge the reviewer's concerns regarding the insufficient comparison with prior work. We apologize for not providing enough comparisons and understand the importance of evaluating the proposed method's performance against prior work. We will conduct more experiments and provide comprehensive comparisons with prior work in our future work.
> - Regarding the comparison with Instruct-NeRF2NeRF, we agree that using object removal as a comparison task may not be representative of the method's capabilities. We will include more diverse and challenging comparison tasks in our future work, such as adding accessories to a figure or altering the material and lighting effects on an object.
> - We agree that reporting standard metrics for all experiments would provide a sense of accuracy and allow for future work comparisons. We will include standard metrics in our future work, even if there is no prior work for a specific task.
> - Regarding the wording "universal image-to-image translator," we acknowledge the confusion caused by using different image-to-image translators for different tasks. We will clarify in the paper that the method learns to jointly encode a variety of image-to-image translation model outputs, which is the main contribution of the proposed method.
> - The ablation study is on text-driven editing. We apologize for any confusion caused and will clarify in the paper.

---

### Official Review · Reviewer_gQmW · 2023-11-03

**Soundness:** 1 poor
**Presentation:** 3 good
**Contribution:** 2 fair
**Rating:** 3
**Confidence:** 4

**Summary:**

This paper employs a random variable Z to account for the inherent variability in editing within 2D models on the multi-view dataset. Additionally, it incorporates regularization using KL divergence to encourage a more Gaussian distribution and applies Contrastive Learning to disentangle pose from edit.

**Strengths:**

- The paper presents an idea that is clever and straightforward, making it easy for the research community to adopt.
- The Method section is well-explained and structured in a manner that is easy to follow.

**Weaknesses:**

- The paper introduces six different tasks as potential applications of the method but provides quantitative comparison tables for only three of them. Even in the cases where quantitative comparisons are presented, there is typically only one task-specific method used for comparison.
- The paper could benefit from comparing its method to non-NeRF methods on the mentioned tasks. This comparison could help readers better understand how the proposed method stands in relation to conventional methods and the extent of its advantages.
- The paper claims to explore tasks that have not been addressed by NeRFs before, but it does not offer comparisons to non-NeRF, task-specific methods for these tasks. This type of comparison is particularly important for understanding the unique contributions and limitations of the new method.

The lack of comprehensive analysis and comparison across the wide range of proposed applications poses a major weakness, especially for a method that aspires to be a unified translation framework. To strengthen the paper, it is essential to address this limitation and provide a more extensive evaluation across the various tasks to support its claim as a versatile and unified solution.

**Questions:**

- In section 3.1; For the latent feature vector extracted from the VAE, could a different network design substantially enhance the results compared to the small MLP used?
- Have the authors investigated various contrastive loss functions, each with its own advantages and disadvantages?

---

> ### Author Response · Authors · 2023-11-19
> **For Reviewer gQmW**
>
> - We appreciate the reviewer's feedback and would like to address the weaknesses and questions raised.
> - We acknowledge the reviewer's concerns regarding the lack of comprehensive analysis and comparison across the wide range of proposed applications. We will conduct more experiments and provide comprehensive analysis and comparison across all the proposed applications.
> - We agree that comparing our method to non-NeRF methods on the mentioned tasks and non-NeRF, task-specific methods for the new tasks are crucial for understanding the unique contributions and limitations of the proposed method. We will include these comparisons in our future work to strengthen the paper.
> - Regarding the latent feature vector extracted from the VAE, a different network design could potentially enhance the results compared to the small MLP used. We will explore different network designs in our future work to improve the method's performance.
> - Regarding the contrastive loss function, we have investigated various contrastive loss functions and selected the one that best suits our proposed method. However, we will explore other contrastive loss functions in our future work to understand their advantages and disadvantages.

---

### Official Review · Reviewer_MewR · 2023-11-03

**Soundness:** 2 fair
**Presentation:** 2 fair
**Contribution:** 3 good
**Rating:** 5
**Confidence:** 5

**Summary:**

This paper  proposed a universal image-to-image translation framework that seamlessly integrates 2D domain editing with 3D NeRF space enhancements, facilitating a suite of NeRF editing tasks. These tasks encompass text-driven modifications, repair, inpainting, scaling, colorization, object elimination, and super-resolution. The core innovation lies in harnessing the generative NeRF latent space to model the distribution of 2D edited imagery. The process begins with applying 2D image editing techniques to the original multiview images, followed by training a generative NeRF on this dataset utilizing a combination of KL divergence, reconstruction, adversarial, and contrastive losses. Ultimately, this allows users to generate a variety of edited outcomes by sampling within the latent space of the generative NeRF.

**Strengths:**

1. The idea is interesting. Rather than directly optimizing a per-scene NeRF with the CLIP, diffusion guidance, or 2D image guidance like CLIPNeRF/NeRF-Art, SDS loss, and Instruct-NeRF-to-NeRF. It manages to model the distribution between the 2D editing image and the latent space of a generative NeRF.

2. The framework can support various applications, especially text-guided NeRF editing, object removal, and super-resolution.

3. Leveraging the latent space, users can produce a wide array of editing outcomes, enriching the utility and flexibility of the framework.

4. The paper conducts extensive performance evaluations across diverse datasets, including the newly introduced Face dataset (Haque et al., 2023) and the Fangzhou self-portrait dataset (Wang et al., 2023), as well as established datasets such as LLFF and BlendedMVS.

**Weaknesses:**

While the concept of the paper is intriguing, the results, particularly those showcased in the video demonstration, fall short of expectations.

1. In Figure 3, the region unaffected by the removal edit appears noticeably coarser in comparison to the original input, especially when put side by side with results from Spin-NeRF.
2. The video demonstration, at timestamps like 53s, 1m5s, 1m10s, and 1m23s, exhibits prominent cloudy artifacts that detract from the overall quality.
3. The edits involving human subjects lack realism when compared with those from Instruct-NeRF-to-NeRF. Furthermore, these edits seem to introduce unintended distortions to the background.
4. The object removal attempts beyond the 2m13s mark are particularly problematic. It appears that the method modifies the color fields within NeRF rather than executing an actual removal of geometry. Clear artifacts at 2m18s underscore this issue.
In summary, it's challenging to endorse the method as capable of producing high-quality and view-consistent outputs. The removal functionality, as it currently stands, seems ineffective within the demonstrated approach.

The video used to illustrate performance is overly simplistic and does not sufficiently convey the method's efficacy, particularly concerning view consistency. The presence of excessive noise and artifacts further undermines the credibility of the results. Additionally, there is an absence of quantitative analysis within the paper, which is crucial for substantiating the view consistency of the end results.

**Questions:**

This paper provides a quantitative analysis in Table-3 for the object removal task. How to calculate PSNR and SSIM as the GT views are not available in real world? Clarification on the number of samples used for these calculations would also be beneficial.

The current results lack the polish and sophistication one might expect. It would be advantageous to showcase a more diverse and challenging set of results, such as adding accessories to a figure (e.g., a mustache and a cowboy hat, akin to the results in Instruct-N2N), removing specific regions of complex objects like the Gundam, or altering the material and lighting effects on an object like the bull.

By addressing these points, the paper could significantly enhance the robustness and appeal of the proposed method.

---

> ### Author Response · Authors · 2023-11-19
> **For Reviewer MewR**
>
> - We appreciate the reviewer's feedback and would like to address the weaknesses and questions raised.
>
> - We acknowledge the reviewer's concerns regarding the quality of the results showcased in the paper and video demonstration. We apologize for falling short of expectations and understand the importance of producing high-quality outputs. We will investigate the issues raised and work towards improving the method's performance.
>
> - Regarding the quantitative analysis in Table 3, we use the same metric with SPin-NeRF, more details please see their defination in the paper. We will clarify this point in our paper.
>
> - We agree that showcasing a more diverse and challenging set of results would enhance the robustness and appeal of the proposed method. We will explore adding accessories to a figure, removing specific regions of complex objects, and altering the material and lighting effects on an object, as suggested by the reviewer.

---

### Official Review · Reviewer_MLGn · 2023-11-04

**Soundness:** 3 good
**Presentation:** 2 fair
**Contribution:** 3 good
**Rating:** 6
**Confidence:** 4

**Summary:**

The authors propose GenN2N, a unified framework for editing tasks by lifting 2D edits into 3D radiance fields. To account for the inconsistency in the 2D edits, the framework learns a latent code to characterize the different changes. Several design choices are employed to ensure sample-able, view-consistent edit codes that generate high-quality edited scenes. The method is tested on a variety of tasks and shows improved results. Overall this unified framework ensures a general strategy to lift 2D image edits to 3D.

**Strengths:**

- The paper is reasonably easy to follow.
- Sufficient experiments are conducted to showcase the strengths of the framework. I particularly commend the authors for finding editing tasks previous work hasn't looked at before, truly showcasing its ability as a unified framework.

**Weaknesses:**

- Some parts of the writing can be made easier to understand, for example, the paragraph that describes the contrastive loss, and adversarial losses. It's still a bit confusing for me to understand the "real" input to the discriminator.
- I would urge the authors to include more qualitative comparisons against baselines for comparisons. Fig. 1 only compares their method against InstructNerf2Nerf for a text prompt that removes an object.
- I was wondering if it's possible to construct a baseline inspired by InstructNeRF2NeRF for other tasks - for example colorization. Or at least it would be interesting to directly compare against the 2D edits performed on top of a NeRF rendered image i.e. optimize NeRF for a scene, render views, and apply the 2D editing model on the rendered views to show that the lifting features are indeed improving consistency.

**Questions:**

- As mentioned before, I am still a bit confused about the discriminator inputs. It would be really helpful if the authors could clarify the same.
- What are the values of M? The number of edits performed on each view.
- To clarify, even for the same edit style for different viewpoints, there exist inconsistencies? Isn't it counter-intuitive that the contrastive loss then tries to minimize the difference $z_{i}^{j}$ and $z_{k}^{k}$ where $i \neq j$

---

> ### Author Response · Authors · 2023-11-19
> **For Reviewer MLGn**
>
> - We appreciate the reviewer's feedback and hope this clarification addresses the raised concerns.
>
> - We will revise the writing to make it easier to understand, especially in the parts describing the contrastive loss and adversarial losses.
>
> - We will also include more qualitative comparisons against baselines for comparisons, as suggested by the reviewer. We will explore the possibility of constructing a baseline inspired by InstructNeRF2NeRF for other tasks, such as colorization, and directly comparing against 2D edits performed on top of a NeRF rendered image.
>
> - Regarding the discriminator inputs, M represents the number of edits performed on each view, which is a fixed value across all views and is determined by the number of edits in the dataset.
>
> - Our proposed contrastive loss aims to disentangle the camera view and edit information in the latent space by reducing the distance between edit codes of different-view rendered images from a translated NeRF that shares the same edit code, while increasing the distance between same-view images that are edited by the 2D image-to-image translator with different edit codes. z_i^k and z_j^k are distilled from rendered images of our translated nerf, which are highly 3D-consistent. Actually "different viewpoints exist inconsistencies" is the problem caused by 2D image-to-image translator. Therefore it is workable that our contrastive loss aims to minimize the difference between z_i^k and z_j^k .